# YOLOv12: Attention-Centric Real-Time Object Detectors

**Yunjie Tian**
University at Buffalo
yunjieti@buffalo.edu

**Qixiang Ye**[*]
UCAS
qxye@ucas.ac.cn

**David Doermann**
University at Buffalo
doermann@buffalo.edu

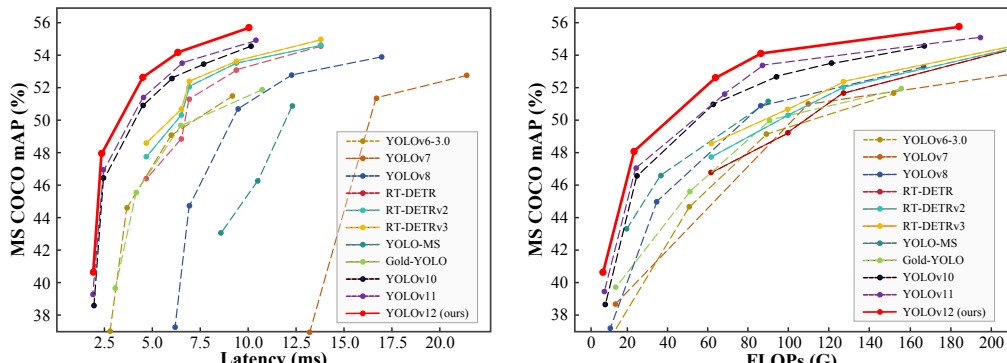

Figure 1: **Comparisons with other popular methods in terms of latency-accuracy (left) and FLOPs-accuracy (right) trade-offs.**

## Abstract

Enhancing the network architecture of the YOLO framework has been crucial for a long time. Still, it has focused on CNN-based improvements despite the proven superiority of attention mechanisms in modeling capabilities. This is because attention-based models cannot match the speed of CNN-based models. This paper proposes an attention-centric YOLO framework, namely YOLOv12, that matches the speed of previous CNN-based ones while harnessing the performance benefits of attention mechanisms. YOLOv12 surpasses popular real-time object detectors in accuracy with competitive speed. For example, YOLOv12-N achieves 40.5% mAP with an inference latency of 1.62 ms on a T4 GPU, outperforming advanced YOLOv10-N / YOLO11-N by 2.0%/1.1% mAP with a comparable speed. This advantage extends to other model scales. YOLOv12 also surpasses end-to-end real-time detectors that improve DETR, such as RT-DETRv2 / RT-DETRv3: YOLOv12-X beats RT-DETRv2-R101 / RT-DETRv3-R101 while running faster with fewer computations and parameters. See more comparisons in Figure 1. Source code is available at https://github.com/sunsmarterjie/yolov12.

Real-time object detection has consistently attracted significant attention due to its low-latency characteristics, which provide substantial practicality [26, 31, 6, 19]. Among them, the YOLO series [51, 53, 52, 5, 32, 35, 64, 26, 67, 61, 31] has effectively established an optimal balance between latency and accuracy, thus dominating the field. Although improvements in YOLO have focused on areas such as loss functions [10, 80, 47, 46, 79, 38, 54], label assignment [25, 37, 68, 24, 81], network architecture design has remained a critical research priority [35, 64, 26, 67, 31]. Although

---

[*]Corresponding author.

39th Conference on Neural Information Processing Systems (NeurIPS 2025).

attention-centric models have been proven to possess more substantial modeling capabilities, even in small models [27, 21, 22, 57], most architectural designs continue to focus primarily on CNNs.

The primary reason for this situation lies in the inefficiency of the attention mechanism, which comes from two main factors: quadratic computational complexity and inefficient memory access operations of the attention mechanism (the latter being the main issue addressed by FlashAttention [16, 15]). As a result, under a similar computational budget, CNN-based architectures outperform attention-based ones by a factor of $\sim 3\times$ [41], which significantly limits the adoption of attention in YOLO systems.

This paper aims to tackle these challenges and establish an attention-centric YOLO framework, YOLOv12. We introduce three key improvements. **First**, we propose a simple yet efficient Area Attention module (A2), which preserves a large receptive field while efficiently reducing the computational complexity of attention, thus improving speed. Moreover, A2 supports flexible input sizes without constraints in window attention [42, 18] to accommodate window partitioning, allowing rectangular inference in YOLO. **Second**, we design Residual Efficient Layer Aggregation Networks (R-ELAN) to address optimization challenges in the designed attention-based models. Building on ELAN [64], R-ELAN introduces (i) a block-level residual design with scaling techniques and (ii) an improved feature aggregation strategy. **Third**, we refine attention-centric architectures to better integrate with the YOLO framework. Key modifications include: incorporating FlashAttention to mitigate memory access issue; using a decoupled projection strategy to construct $q$, $k$, and $v$ during attention computation, thus avoiding redundant feature reorganization; removing positional encoding for a leaner design; reducing the MLP ratio from 4 to 1.5 to better balance attention and FFN computation; and decreasing the depth of stacked blocks to facilitate optimization.

Based on the designs outlined above, we develop a new family of real-time detectors with 5 model scales: YOLOv12-N, S, M, L, and X. We perform extensive experiments on standard object detection benchmarks following YOLO11 [31] without any additional tricks, demonstrating that YOLOv12 provides significant improvements over previous popular models in terms of latency-accuracy and FLOPs-accuracy trade-offs across these scales, as illustrated in Figure 1. For example, YOLOv12-N achieves 40.5% mAP, outperforming YOLOv10-N [61] by 2.0% mAP while maintaining a faster inference speed, and YOLO11-N [31] by 1.1% mAP with a comparable speed. This advantage remains consistent across other scale models. Compared to RT-DETRv2-R18 [78], YOLOv12-S is comparable and 47% faster in latency speed, requiring only 33% of its computations and 46% of its parameters. Compared to RT-DETRv3-R50 [43], YOLOv12-L is 0.4% mAP better and 15% faster, requiring only 61% of its computations and 63% of its parameters.

In summary, YOLOv12 contributes by: **(i)** introducing an attention-centric, efficient YOLO framework that challenges CNN dominance in the series, and **(ii)** achieving state-of-the-art results with fast inference and high accuracy without relying on pre-training or additional training techniques.

# 1   Related Work

**Real-time Object Detectors.** The YOLO series [51, 53, 52, 5, 32, 35, 64, 63, 11, 26, 67, 61, 31] has become the leading framework for real-time object detection. The early YOLO models [51, 53, 52] established the foundation primarily from a model design perspective. YOLOv4 [5] and YOLOv5 [32] introduced CSPNet [65], data augmentation, and multi-scale features, while YOLOv6 [35] further enhanced efficiency with BiC, SimCSPSPPF, *etc*. YOLOv7 [64] incorporated E-ELAN [66] for better gradient flow and various bag-of-freebies, while YOLOv8 [26] adopted the efficient C2f block for feature extraction. Recent versions, YOLOv9 [67], introduced GELAN for architectural optimization and PGI for training improvements, YOLOv10 [61] applied NMS-free training with dual assignments, and YOLOv11 [31] optimized speed and accuracy with C3K2 (a variant of GELAN) and lightweight depthwise separable convolutions. Other developments [73, 62, 75] further enhanced detection capabilities through improved backbones and head designs Beyond YOLO, RT-DETR [78] improved end-to-end detectors [9, 83, 45, 36] for real-time use through an efficient encoder and uncertainty-minimal query selection, with RT-DETRv2 [43] and RT-DETRv3 [69] further refinement with bag-of-freebies. Recent follow-ups [50, 29] continue to show compromising results.

**Efficient Vision Transformers.** Reducing the computational cost of global self-attention is the key to effectively applying vision transformers to downstream tasks. PVT [71] tackles this with multi-resolution stages and downsampling. Swin Transformer [42] restricts self-attention to local windows and shifts them to connect non-overlapping regions. Other approaches, like axial self-attention [28]

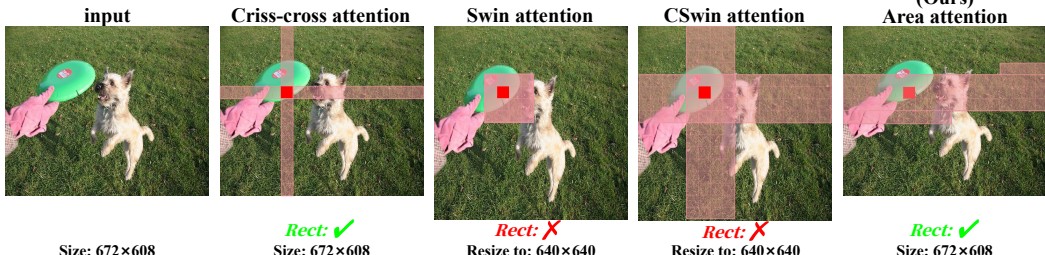

Figure 2: **Comparison of the representative local attention mechanisms with our area attention.** Area attention computes local attention over sequentially placed tokens without requiring input size constraints, enabling default distortion-free rectangular inference (*Rect*) in YOLO. Area attention enjoys efficient implementation, requiring only `flatten` and a `reshape` operations.

and criss-cross attention [30], compute attention within horizontal and vertical windows. CSWin Transformer [18] extends this with self-attention in cross-shaped windows. Furthermore, methods such as [14, 76] establish local-global relations, enhancing efficiency while reducing the reliance on global self-attention. Fast-iTPN [57] accelerates downstream task inference with token migration and gathering. Some methods [56, 70, 33] use linear attention to reduce complexity, but vision mamba models [82, 41] still struggle to achieve real-time speeds [41]. FlashAttention [16, 15] addresses high-bandwidth memory bottlenecks by optimizing I/O and reducing memory access for efficiency.

## 2 Approach

### 2.1 Efficiency Analysis

The attention mechanism, while highly effective in capturing global dependencies and facilitating tasks such as natural language processing [7, 17] and computer vision [23, 42], is inherently slower than CNNs. Two primary factors contribute to this speed discrepancy.

**Complexity.** First, the computational complexity of the attention operation scales quadratically with the input sequence length $L$. Specifically, for an input sequence with length $L$ and feature dimension $d$, the computation of the attention matrix requires $O(L^2d)$ operations since each token attends to every other token. In contrast, the complexity of convolution operations in CNNs scales linearly with respect to the spatial or temporal dimension, *i.e.*, $O(kLd)$, where $k$ is the kernel size and is typically much smaller than $L$. As a result, self-attention becomes computationally prohibitive, especially for significant inputs such as high-resolution images or long sequences.

Moreover, attention-based vision transformers often suffer from speed overhead due to their complex designs (*e.g.*, window partitioning / reverse in some window attentions [42, 18]) and additional modules (*e.g.*, positional encoding), leading to slower performance compared to CNNs [41]. In contrast, the design modules in this paper use simple and efficient operations to implement attention, maximizing speed and efficiency.

**Computation.** Second, the memory access patterns in attention computations are less efficient than in CNNs [16, 15]. In self-attention, intermediate maps such as the attention map ($QK^T$) and the softmax map ($L \times L$) must be transferred from high-speed GPU SRAM to high-bandwidth GPU memory (HBM) for further computation. The read/write speed of SRAM is more than 10 times faster, leading to significant memory access overhead and increased wall clock time.

### 2.2 Area Attention

A straightforward way to reduce the computational cost of vanilla attention is to use the linear attention mechanism [56, 70], which reduces the complexity from quadratic to linear. For a visual feature $f$ with dimensions $(n, h, d)$, where $n$ is the number of tokens, $h$ is the number of heads, and $d$ is the head size, linear attention reduces the complexity from $2n^2hd$ to $2nhd^2$, cutting the computational cost as $n > d$. However, linear attention faces issues such as global dependency degradation [34],

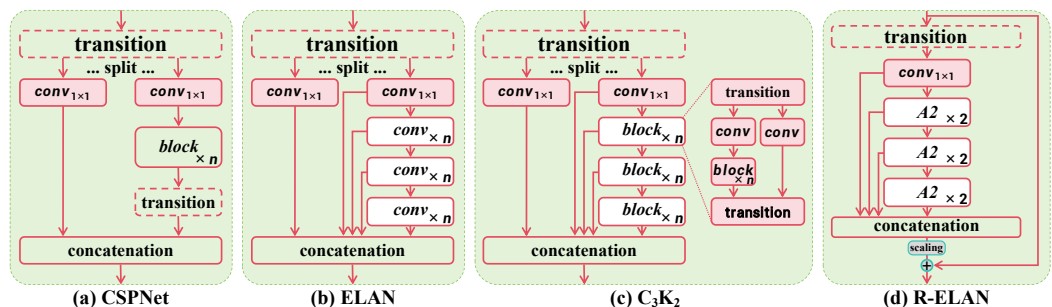

Figure 3: **The architecture comparison with popular modules** including (a): CSPNet [65], (b) ELAN [66], (c) C3K2 (a case of GELAN) [67, 31], and (d) the proposed R-ELAN (residual efficient layer aggregation networks).

instability [12], and distribution sensitivity [72]. Furthermore, the low-rank bottleneck [13, 3] provides limited speed improvements when applied to YOLO.

An alternative approach to effectively reduce complexity is the local attention mechanism (*e.g.*, criss-cross attention [30], axial attention [28], and window attention [42, 18]), as shown in Figure 2, which transforms global attention into local, thus reducing computational costs. However, partitioning the feature map into windows can introduce overhead or reduce the receptive field, impacting both speed and accuracy. Furthermore, for window attentions, such as Swin [42] and CSwin [18], they also suffer from additional constraints on input size to accommodate window partitioning. As shown in Figure 2, they cannot handle specific rectangular dimensions (native resolutions) commonly seen in the inference of YOLO (*e.g.*, $672 \times 608$) and must resize the input to window-friendly sizes (*e.g.*, $640 \times 640$), which somehow introduces image distortion and thereby degrades detection performance.

In this study, we propose a simple yet efficient area attention module (A2). The comparison of A2 with other local attentions is illustrated in Figure 2. Specifically, for a feature of size $(H, W)$, we first perform a flatten operation to obtain a one-dimensional feature with $H \times W$ tokens, which is then evenly divided into $l$ segments (areas) (so each area contains $\frac{H \times W}{l}$ sequentially placed tokens). Local attention is applied independently within each area (a simple reshape operation enables the transformation to the original size). This only requires a simple flatten and a reshape operation, leading to efficient implementation and faster speed. We empirically set the default value of $l$ to 4 (or 8), thus reducing the receptive field to $\frac{1}{l}$ of the original size while maintaining a large receptive field. More importantly, A2 eliminates feature size constraints while only requiring the total token count $(H \times W)$ to be divisible by $l$. This design inherently supports rectangular inference (see Figure 2), the standard evaluation protocol in YOLO, thus achieving seamless compatibility with the YOLO framework. (See more details about A2 and architecture design in the Appendix.)

Regarding the memory access issue, we integrate FlashAttention [16, 15] into area attention to solve it. FlashAttention is already an infrastructure for many large language models [60, 1, 40] and diffusion models [55, 49, 4], and can provide similar benefits to YOLO.

## 2.3 Residual Efficient Layer Aggregation Networks

Efficient layer aggregation networks (ELAN) [64] are designed to improve feature aggregation. As shown in Figure 3 (b), ELAN splits the output of a transition layer (a convolution $1 \times 1$, processes each split through multiple modules, and then concatenates the outputs before applying another transition layer to align the dimensions. However, as noted in [64], this design can lead to instability. We argue that this architecture causes gradient blocking and lacks residual connections between input and output. In addition, incorporating the attention mechanism into the network introduces additional optimization challenges. Empirically, L- and X-scale models either fail to converge or remain unstable, even when using Adam or AdamW optimizers.

To address this issue, we propose residual efficient layer aggregation networks (R-ELAN), as shown in Figure 3 (d). We introduce a residual shortcut from input to output throughout the block, with a scaling factor (defaulting to 0.01). This design is inspired by the layer scaling [59] used in deep vision transformers. However, applying layer scaling for each attention area does not solve the optimization

Table 1: **Comparison with popular real-time object detectors.** All results are obtained using $640 \times 640$ inputs. †: pre-trained models are required.

| Method | FLOPs (G) | #Param. (M) | $\text{AP}^{val}_{50:95}$ (%) | $\text{AP}^{val}_{50}$ (%) | $\text{AP}^{val}_{75}$ (%) | Latency (ms) |
|---|---|---|---|---|---|---|
| YOLOv8-N [26] | 8.7 | 3.2 | 37.4 | 52.6 | 40.5 | 1.77 |
| YOLOv10-N [61] | 6.7 | 2.3 | 38.5 | 53.8 | 41.7 | 1.84 |
| YOLO11-N [31] | 6.5 | 2.6 | 39.4 | 55.3 | 42.8 | 1.5 |
| **YOLOv12-N (Ours)** | **6.0** | **2.6** | **40.5** | **56.4** | **43.8** | **1.62** |
| YOLOv8-S [26] | 28.6 | 11.2 | 45.0 | 61.8 | 48.7 | 2.33 |
| RT-DETRv2-R18† [44] | 60.0 | 20.0 | 47.9 | 64.9 | – | 4.58 |
| YOLOv9-S [67] | 26.4 | 7.1 | 46.8 | 63.4 | 50.7 | – |
| YOLOv10-S [61] | 21.6 | 7.2 | 46.3 | 63.0 | 50.4 | 2.49 |
| YOLO11-S [31] | 21.5 | 9.4 | 46.9 | 63.9 | 50.6 | 2.5 |
| **YOLOv12-S (Ours)** | **19.5** | **9.1** | **47.8** | **64.9** | **51.3** | **2.44** |
| YOLOv8-M [26] | 78.9 | 25.9 | 50.3 | 67.2 | 54.7 | 5.09 |
| RT-DETRv2-R34† [44] | 100.0 | 36.0 | 49.9 | 67.5 | – | 6.32 |
| RT-DETRv3-R18† [44] | 60.0 | 20.0 | 48.7 | – | – | 4.58 |
| RT-DETRv3-R34† [69] | 100.0 | 36.0 | 50.1 | 67.5 | – | 6.32 |
| YOLOv9-M [67] | 76.3 | 20.0 | 51.4 | 68.1 | 56.1 | – |
| YOLOv10-M [61] | 59.1 | 15.4 | 51.1 | 68.1 | 55.8 | 4.74 |
| YOLO11-M [31] | 68.0 | 20.1 | 51.5 | 68.5 | 55.7 | 4.7 |
| **YOLOv12-M (Ours)** | **59.9** | **19.7** | **52.5** | **70.0** | **57.0** | **4.30** |
| YOLOv8-L [26] | 165.2 | 43.7 | 53.0 | 69.8 | 57.7 | 8.06 |
| RT-DETRv2-R50† [44] | 136.0 | 42.0 | 53.4 | 71.6 | – | 6.90 |
| RT-DETRv3-R50† [69] | 136.0 | 42.0 | 53.4 | – | – | 6.90 |
| YOLOv9-C [67] | 102.1 | 25.3 | 53.0 | 70.2 | 57.8 | – |
| YOLOv10-B [61] | 92.0 | 19.1 | 52.5 | 69.6 | 57.2 | 5.74 |
| YOLOv10-L [61] | 120.3 | 24.4 | 53.2 | 70.1 | 58.1 | 7.28 |
| YOLO11-L [31] | 86.9 | 25.3 | 53.3 | 70.1 | 58.2 | 6.2 |
| D-FINE-L† [50] | 91 | 31 | 54.0 | 71.6 | 58.4 | 8.07 |
| **YOLOv12-L (Ours)** | **82.6** | **26.6** | **53.8** | **71.1** | **58.7** | **5.89** |
| YOLOv8-X [26] | 257.8 | 68.2 | 54.0 | 71.0 | 58.8 | 12.83 |
| RT-DETRv2-R101† [44] | 259.0 | 76.0 | 54.3 | 72.8 | – | 13.5 |
| RT-DETRv3-R101† [69] | 259.0 | 76.0 | 54.6 | – | – | 13.5 |
| YOLOv10-X [61] | 160.4 | 29.5 | 54.4 | 71.3 | 59.3 | 10.70 |
| YOLO11-X [31] | 194.9 | 56.9 | 54.6 | 71.6 | 59.5 | 11.3 |
| D-FINE-X† [50] | 202 | 62 | 55.8 | 73.7 | 60.2 | 12.89 |
| **YOLOv12-X (Ours)** | **184.9** | **59.5** | **55.4** | **72.6** | **60.4** | **10.47** |

challenge and introduces latency. This highlights that the convergence issue is not solely due to the attention mechanism but also the ELAN structure, validating the rationale behind R-ELAN design.

We also design a new feature aggregation approach, shown in Figure 3 (d). In the original ELAN, the input is passed through a transition layer, splitting it into two parts. Subsequent blocks process one part, and both parts are concatenated to produce the output. In contrast, our design uses a transition layer to adjust channel dimensions and produce a single feature map. This map is processed through subsequent blocks, followed by concatenation, forming a bottleneck structure. This method retains the original feature integration capability while reducing computational cost, parameters, and memory usage.

## 2.4 Architectural Improvements

Many attention-based vision transformers use plain-style architectures [20, 58, 2, 27, 23, 22], while we retain the hierarchical structure of previous YOLO versions [51, 53, 52, 5, 32, 35, 64, 26, 67, 61, 31]

and we will demonstrate its necessity. We simplify the architecture depth by removing the stacking of three blocks in the final stage of the backbone that are used most frequently in recent YOLO versions [26, 67, 61, 31], retaining only a single block. In addition, we retain the first two blocks, remove the third block, and replace all C3K2 blocks with R-ELAN blocks in the backbone. We used convolutions with the 2 and 4 groups in the second and third blocks, respectively.

Several default configurations of the vanilla attention mechanism have also been modified for better alignment with the YOLO system. These include adjusting the MLP ratio from 4 to 1.5 (or 2 for N / S / M scale models), removing positional encoding, and adding a separable convolution ($7 \times 7$) (position perceiver) to enhance the ability of area attention to perceive positional information. The effectiveness of these changes will be demonstrated in Section 3.5.

Previous versions (*e.g.*, YOLOv10 [61] and YOLOv11 [31]) adopt a coupled projection strategy for constructing $q$, $k$, and $v$ during attention calculation, where a single convolutional layer jointly projects the input and then splits the output into the three components. However, when used alongside the position perceiver, this coupling results in redundant reorganization of the $v$ features, degrading inference efficiency. To mitigate this, we introduce a decoupled projection strategy that computes $v$ separately from $q$ and $k$. This design eliminates unnecessary processing on $v$, leading to an inference speedup of approximately 10%.

Table 2: **Comparison of YOLOv12 with previous versions in speed (GPU, CPU) and memory usage.** CUDA results are measured on T4 / RTX 3080 GPUs, with inference latency (ms) reported for FP32 and FP16. Memory usage (Mem.) is measured with the TensorRT model (bs = 1). All results are obtained using the same hardware.

| Model | FLOPs (G) | Mem. (G) | CUDA FP32 | CUDA FP16 | CPU | mAP |
|---|---|---|---|---|---|---|
| YOLOv9-T [67] | 8.2 | 0.16 | 4.0/2.3 | 2.4/1.5 | 40.1 | – |
| YOLOv10-N [61] | 6.7 | 0.19 | 3.0/1.6 | 1.7/1.0 | 32.4 | 38.5 |
| YOLO11-N [31] | 6.5 | 0.20 | 3.0/1.6 | 1.5/0.9 | 32.5 | 39.5 |
| YOLOv12-N | 6.0 | 0.15 | 3.0/1.6 | 1.6/1.0 | 38.7 | 40.5 |
| YOLOv9-S [67] | 26.4 | 0.19 | 7.3/3.7 | 3.3/1.9 | 85.6 | 46.8 |
| YOLOv10-S [61] | 21.6 | 0.23 | 6.3/2.6 | 2.6/1.3 | 70.1 | 46.3 |
| YOLO11-S [31] | 21.5 | 0.23 | 6.4/2.7 | 2.4/1.3 | 72.1 | 46.9 |
| YOLOv12-S | 19.5 | 0.18 | 6.5/2.7 | 2.5/1.3 | 83.3 | 47.8 |
| YOLOv9-M [67] | 76.3 | 0.24 | 16.7/6.3 | 5.6/2.7 | 186.5 | 51.4 |
| YOLOv10-M [61] | 59.1 | 0.29 | 13.5/5.3 | 4.8/2.4 | 161.5 | 51.1 |
| YOLO11-M [31] | 68.0 | 0.30 | 16.2/5.4 | 4.4/2.2 | 189.9 | 51.5 |
| YOLOv12-M | 59.9 | 0.24 | 14.9/5.2 | 4.3/2.1 | 213.2 | 52.5 |
| YOLOv9-C [67] | 102.1 | 0.29 | 20.9/7.4 | 6.2/2.8 | 272.6 | 53.0 |
| YOLOv10-L [61] | 120.3 | 0.30 | 23.8/8.0 | 7.3/3.4 | 294.8 | 53.2 |
| YOLO11-L [31] | 86.9 | 0.32 | 20.4/6.9 | 5.9/2.9 | 237.7 | 53.3 |
| YOLOv12-L | 82.6 | 0.27 | 20.6/6.9 | 6.0/2.9 | 299.5 | 53.8 |
| YOLOv9-E [67] | 189.0 | 0.69 | 48.6/15.5 | 15.5/6.5 | 499.7 | 55.6 |
| YOLOv10-X [61] | 160.4 | 0.40 | 35.0/10.7 | 10.4/4.5 | 410.1 | 54.4 |
| YOLO11-X [31] | 194.9 | 0.45 | 40.4/13.5 | 10.5/4.9 | 484.0 | 54.6 |
| YOLOv12-X | 184.9 | 0.37 | 40.6/13.6 | 10.5/4.9 | 524.6 | 55.4 |

## 3 Experiment

### 3.1 Experimental Setup

We validate YOLOv12 on the MS-COCO 2017 dataset [39]. The YOLOv12 family consists of 5 variants: YOLOv12-N, YOLOv12-S, YOLOv12-M, YOLOv12-L, and YOLOv12-X. All models are trained for 600 epochs using the SGD optimizer with an initial learning rate of 0.01, consistent with YOLO11 [31]. A linear learning rate decay schedule is adopted, with a linear warm-up for the first three epochs. Following the methodology in [61, 78], the latencies of all models are tested on a T4 GPU using TensorRT FP16. See the Appendix for more details and results.

### 3.2 Comparison with State-of-the-arts

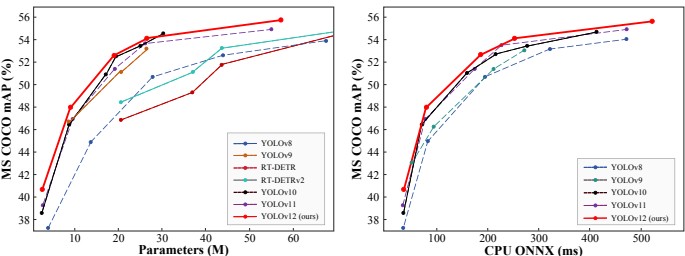

Figure 4: **Accuracy-parameter/CPU latency trade-offs for YOLOv12.**

We compare YOLOv12 with other detectors in Table 1, using performance data from their official reports (some achieve higher speed in our reproduction, see Table 2). **N-scale**: YOLOv12-N exceeds YOLOv8-N [67], YOLOv10-N [61], and YOLO11 [31] by up to 3.1% mAP, maintaining a similar or lower computational cost with 1.62 ms/image latency. **S-scale**: YOLOv12-S (19.5G FLOPs, 9.1M params) achieves 47.8 mAP

at 2.44 ms/image, outperforming YOLOv9-S [67], YOLOv10-S [61], and YOLO11-S [31]. Compared to RT-DETRv2-R18 [44], it offers better speed with lower computational cost. **M-scale**: YOLOv12-M (59.9G FLOPs, 19.7M params) achieves 52.5 mAP at 4.30 ms/image, outperforming YOLOv9-M [67], YOLOv10 [61], YOLO11 [31], and RT-DETRv2-R34 [44] / RT-DETRv3-R34 [69]. **L-scale**: YOLOv12-L exceeds YOLOv10-L [61] with 37.7G fewer FLOPs and YOLO11-L [31] by 0.5% mAP while maintaining a similar efficiency. It also outperforms RT-DETRv2-R50 [44] / RT-DETRv3-R50 [69] with fewer FLOPs, parameters, and faster speed. **X-scale**: YOLOv12-X exceeds YOLOv10-X [61] and YOLO11-X [31] by up to 1.0% mAP with similar efficiency and surpasses RT-DETRv2-R101 [43] / RT-DETRv3-R101 [69] with 28.6% fewer FLOPs, 21.7% fewer parameters, and 22.4% faster speed.

## 3.3 Speed, Memory, and Efficiency

Table 2 compares the inference speed of YOLOv9 [67], YOLOv10 [61], YOLO11 [31], and YOLOv12 on T4 and RTX 3080 GPUs using FP32 and FP16 precision, as well as CPU (Intel Core i7-10700K @ 3.80GHz) speed and peak memory usage. All results are obtained on the same hardware to ensure fairness. The results show that YOLOv12 significantly outperforms YOLOv9 in inference speed, while matching the performance of YOLOv10 and YOLOv11. YOLOv12 requires less peak memory than the other models.

Figure 4 compares the speed and efficiency of YOLOv12. We present the accuracy-parameter trade-off, demonstrating that it outperforms popular methods, including YOLOv10, which has significantly fewer parameters.

Table 3: **Ablation on the proposed area attention.** Compared to other methods (cross-cross (CC), Swin, and CSwin attentions), area attention helps YOLOv12-N/S/X models obtain better accuracy and run faster on GPU (CUDA) and CPU. CUDA results are measured on T4/RTX 3080. All results use the same hardware and exclude FlashAttention [16, 15].

| Model | Atten. Type | Mem. (G) | CUDA (ms) FP32 | CUDA (ms) FP16 | CPU (ms) | $AP_{50:95}^{val}$ |
|---|---|---|---|---|---|---|
| N | Global | 0.21 | 6.1/2.7 | 3.1/1.6 | 66.3 | 41.7 |
|   | CC | 0.17 | 4.7/2.3 | 2.4/1.4 | 72.9 | 39.5 |
|   | Swin | 0.16 | 4.5/2.2 | 2.2/1.3 | 39.6 | 40.2 |
|   | CSwin | 0.16 | 4.5/2.2 | 2.2/1.3 | 39.9 | 40.3 |
|   | A2 | 0.15 | 4.2/2.0 | 2.0/1.2 | 38.7 | 40.5 |
| S | Global | 0.30 | 12.5/4.8 | 5.2/2.4 | 144.3 | 49.0 |
|   | CC | 0.20 | 9.5/3.9 | 3.8/2.1 | 90.6 | 46.6 |
|   | Swin | 0.18 | 9.2/3.8 | 3.3/1.9 | 83.4 | 47.4 |
|   | CSwin | 0.18 | 9.3/3.8 | 3.3/1.9 | 85.6 | 47.5 |
|   | A2 | 0.18 | 8.7/3.5 | 3.1/1.8 | 83.3 | 47.8 |
| X | Global | 0.68 | 74.2/31.4 | 26.3/10.8 | 913.0 | 56.2 |
|   | CC | 0.40 | 56.2/23.6 | 16.6/8.0 | 572.3 | 54.7 |
|   | Swin | 0.38 | 53.4/21.2 | 15.3/7.6 | 538.2 | 55.2 |
|   | CSwin | 0.38 | 53.5/21.4 | 15.4/7.6 | 553.6 | 55.2 |
|   | A2 | 0.37 | 52.5/21.0 | 14.7/7.1 | 524.6 | 55.4 |

This underscores YOLOv12's ability to achieve higher accuracy with fewer parameters. We also include the accuracy-latency trade-off for YOLOv12 on a CPU (Intel Core i7-10700K @ 3.80GHz). As shown, YOLOv12 delivers superior performance and establishes superior boundaries, demonstrating greater efficiency across various metrics.

## 3.4 Ablation Studies

We perform ablation experiments to assess the effectiveness of area attention (A2) and E-ELAN in Table 3 and Table 4, respectively.

• **Area Attention.** In Table 3, evaluations are performed on YOLOv12-N/S/X models, measuring the GPU (CUDA) and CPU inference speed. CUDA results are obtained using identical T4 and RTX 3080 hardware to ensure fairness, while CPU performance is measured on an Intel Core i7-10700K @ 3.80GHz. Memory usage (Mem.) is measured with the TensorRT model (bs = 1). We compare the performance with other local attentions and adjust the number of chan-

Table 4: **Ablation on the proposed residual efficient layer aggregation networks (R-ELAN).** RA: Proposed feature integration; RS: Residual block; SL: Scaling factor for residuals; CS: Convergence status.

| Model scale | RA | RS | SL | CS | FLOPs (G) | Param (M) | Mem. (G) | mAP | Lat. |
|---|---|---|---|---|---|---|---|---|---|
| N | ✗ | ✗ | – | ✔ | 6.5 | 2.8 | 0.20 | 40.6 | 1.70 |
|   | ✔ | ✗ | – | ✔ | 6.0 | 2.6 | 0.15 | 40.4 | 1.62 |
|   | ✔ | ✔ | 0.01 | ✔ | 6.0 | 2.6 | 0.15 | 40.5 | 1.62 |
| L | ✗ | ✗ | – | ✗ | – | – | – | – | – |
|   | ✔ | ✗ | – | ✗ | – | – | – | – | – |
|   | ✗ | ✔ | 0.01 | ✔ | 87.7 | 27.2 | 0.31 | 53.9 | 6.15 |
|   | ✔ | ✔ | 0.1 | ✔ | 82.6 | 26.6 | 0.27 | 53.7 | 5.89 |
|   | ✔ | ✔ | 0.01 | ✔ | 82.6 | 26.6 | 0.27 | 53.8 | 5.89 |
| X | ✗ | ✗ | – | ✗ | – | – | – | – | – |
|   | ✔ | ✗ | – | ✗ | – | – | – | – | – |
|   | ✗ | ✔ | 0.01 | ✔ | 197.5 | 60.8 | 0.42 | 55.4 | 10.47 |
|   | ✔ | ✔ | 0.1 | ✔ | 184.9 | 59.5 | 0.37 | 55.2 | 10.47 |
|   | ✔ | ✔ | 0.01 | ✔ | 184.9 | 59.5 | 0.37 | 55.4 | 10.47 |

Table 5: **Diagnostic studies.** To save space, we only show the factor(s) to be diagnosed in each subtable. The default parameters are (unless otherwise specified) training for 600 epochs from scratch, using the YOLOv12-N model. All latency (Lat.) results are tested on a T4 GPU

| Model | Lat.$^{-DP}$ | Lat. |
|---|---|---|
| N | 1.70 | 1.62 |
| S | 2.70 | 2.44 |
| M | 4.88 | 4.30 |
| L | 6.56 | 5.89 |
| X | 11.39 | 10.47 |

(a) Decoupled Projection

| Method | mAP | Lat. |
|---|---|---|
| N/A | 38.2 | 1.60 |
| $S_T$ | 40.1 | 1.60 |
| $S_4$ | 39.8 | 1.68 |
| Ours | 40.5 | 1.62 |

(b) Hierarchical Design

| Ep. | mAP (N) | mAP (S) |
|---|---|---|
| 300 | 39.2 | 47.0 |
| 500 | 40.3 | 47.6 |
| 600 | 40.5 | 47.8 |
| 800 | 41.1 | 48.1 |

(c) Training Epoch

| kernel | mAP | Lat. |
|---|---|---|
| 3 | 40.1 | 1.57 |
| 5 | 40.4 | 1.60 |
| 7 | 40.5 | 1.62 |
| 9 | 40.6 | 1.73 |

(d) Position Perceiver

| Pos. | mAP | Lat. |
|---|---|---|
| RPE | 40.2 | 1.77 |
| APE | 40.3 | 1.67 |
| N/A | 40.5 | 1.62 |

(e) Position Embedding

| Model | mAP | Lat. |
|---|---|---|
| YOLOv10 | 38.5 | 1.68 |
| YOLO11 | 39.4 | 1.49 |
| YOLOv12 | 40.5 | 1.62 |

(f) FA for V10/11

| Ratio (L) | mAP | Lat. |
|---|---|---|
| 1.5 | 53.8 | 5.89 |
| 2.0 | 53.3 | 5.79 |
| 4.0 | 53.1 | 5.73 |

(g) MLP Ratio

| FA | Lat. (N) | Lat. (S) |
|---|---|---|
| ✗ | 2.00 | 3.12 |
| ✔ | 1.62 | 2.45 |

(h) FlashAttention

nels to ensure that all models have similar parameters and FLOPs. The results demonstrate significant efficiency and speedup with area attention. For example, with FP32 on RTX 3080, YOLOv12-N achieves a 1.2ms inference (RTX 3080) with a 40.5 mAP, surpassing the criss-cross (CC) [30], Swin [42], and CSwin [18] attentions, with only 0.15G inference memory usage. In particular, the Swin and CSwin attentions require a resize operation (to $640 \times 640$) during inference, which can cause a performance drop[2]. Performance gain is consistently observed across different models and hardware configurations. We do not use FlashAttention [16, 15] in this experiment because it would significantly reduce the speed difference.

• **R-ELAN.** In Table 4, evaluations are performed on YOLOv12-N/L/X models, revealing two key findings: **(i)** For smaller models such as YOLOv12-N, residual connections provide performance gain with negligible extra cost. For larger models (YOLOv12-L/X), residual connections are crucial for stable training, with a minimal scaling factor (0.01) for convergence. **(ii)** The proposed feature integration method reduces the complexity of the model (FLOPs and parameters) and the memory cost while maintaining accuracy, with only a slight (even without) performance drop.

### 3.5 Diagnosis & Visualization

We diagnose YOLOv12 designs in Tables 5a to 5h, using YOLOv12-N trained from scratch for 600 epochs, unless otherwise stated.

• **Decoupled Projection: Table 5a.** We compare the Decoupled Projection (DP) strategy to construct $q$, $k$, and $v$ with previous methods (Lat.$^{-DP}$). DP consistently improves speed by around 10% in all scale models, as it avoids costly reorganization of the $v$ feature during the A2 implementation, leading to better efficiency.

• **Hierarchical Design: Table 5b.** Unlike other detection systems, such as Mask R-CNN [27, 2], where plain vision transformers produce substantial results, YOLOv12 behaves differently. Using a plain vision transformer (N/A) causes a significant performance drop, achieving only 38.2% mAP. A moderate adjustment, such as omitting the first ($S_T$) or fourth stage ($S_4$) while maintaining similar FLOPs, results in a slight performance degradation of 0.4% mAP and 0.7% mAP, respectively. Consistent with previous YOLO models, the hierarchical design remains the most effective, providing the best performance in YOLOv12.

• **Training Epochs: Table 5c.** We examine how varying the number of training epochs impacts performance (training from scratch). Although some existing YOLO detectors achieve optimal results after roughly 500 training epochs [26, 67, 61], YOLOv12 requires a more extended training

---

[2]If we resize the image to $640 \times 640$ for A2 inference, the speed remains unaffected, but the performance drops by about 0.2 in general, highlighting the importance of rectangular inference.

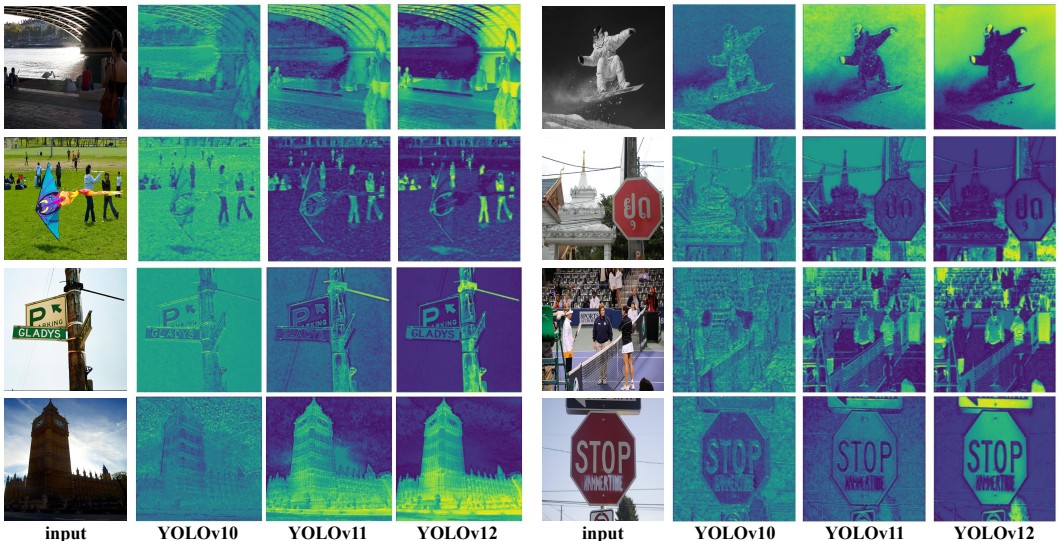

| input | YOLOv10 | YOLOv11 | YOLOv12 | input | YOLOv10 | YOLOv11 | YOLOv12 |

Figure 5: **Comparison of heat maps between YOLOv10 [61], YOLO11 [31], and the proposed YOLOv12.** YOLOv12 exhibits clearer object perception than YOLOv10 and YOLO11. All results use X-scale models. *Zoom in to compare details.*

period (about 600/800 epochs) to achieve peak performance, keeping the same configuration used in YOLO11 [31].

• **Position Perceiver: Tables 5d.** In the attention mechanism, we apply a separable convolution with a large kernel to the attention value $v$, adding its output to $v$@attn. This component, called the Position Perceiver, preserves the original positions of image pixels, helping the attention mechanism perceive positional information. While similar to the PSA module [61], we expand the convolution kernel, improving performance without affecting speed. Increasing the kernel size enhances performance but slows processing, with a significant slowdown at $9 \times 9$. Therefore, we set the default kernel size to $7 \times 7$.

• **Position Embedding: Tables 5e.** We examine the impact of common positional embeddings (RPE and APE) on performance. Interestingly, the best results are achieved without any positional embedding, leading to a cleaner architecture and faster inference.

• **FA for V10/11: Tables 5f.** This table uses the FlashAttention (FA) for YOLOv10 and YOLO11, including a few attention blocks. It can be seen that with FA, they also benefit from speed improvements. FA should serve as future infrastructure for the YOLO framework, much like in large language models.

• **MLP Ratio: Tables 5g.** In vanilla attention, the MLP ratio within the attention module is typically set to 4.0. However, YOLOv12 behaves differently. The table shows that varying the MLP ratio affects model size, so we adjust the feature dimensions for consistency. In particular, YOLOv12 performs better with an MLP ratio of 1.5, shifting the computational load toward the attention mechanism and emphasizing the importance of area attention.

• **FlashAttention: Tables 5h.** This table demonstrates the role of FlashAttention in YOLOv12, showing a 0.38ms speedup for YOLOv12-N and 0.67 ms for YOLOv12-S without additional costs.

**Visualization: Heat Map Comparison.** Figure 5 compares the heat maps of YOLOv12 with YOLOv10 [61] and YOLO11 [31]. These heat maps, extracted from the third stage of the backbones of X-scale models, highlight the activated regions, reflecting the model's object perception capability. As illustrated, YOLOv12 shows more defined object contours and better foreground activation than YOLOv10 and YOLO11, indicating improved perception. We explain that this improvement comes from the area attention mechanism, which captures a broader context and enables more precise foreground activation with its larger receptive field than CNN with its larger receptive field than CNN. We believe that this characteristic gives YOLOv12 a performance advantage.

# 4 Conclusion

This study introduces YOLOv12, which integrates an attention-centric design into the YOLO framework, achieving a state-of-the-art latency-accuracy trade-off. We propose a novel network that uses area attention to reduce computational complexity and R-ELAN to enhance feature aggregation for efficient inference. Key refinements to the vanilla attention mechanism further align it with YOLO's real-time constraints while maintaining high-speed performance. By combining area attention, R-ELAN, and architectural optimizations, YOLOv12 significantly improves accuracy and efficiency. Comprehensive ablation studies validate these innovations. This work challenges CNN-dominated YOLO designs and advances attention-based real-time object detection.

**Limitations.** YOLOv12 requires FlashAttention [16, 15], which currently supports Turing, Ampere, Ada Lovelace, or Hopper GPUs (*e.g.*, T4, Quadro RTX series, RTX20 series, RTX30 series, RTX40 series, RTX A5000/6000, A30/40, A100, H100, *etc.*).

# 5 Acknowledgment

This work was supported by National Natural Science Foundation of China (NSFC) under Grant 62225208 and 62450046 and CAS Project for CAS Project for Young Scientists in Basic Research under Grant No.YSBR-117.

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

Table 6: **Hyperparameters for training the YOLOv12 family on COCO [39].**

| Hyperparameters | N/S/M/L/X-Scale |
|---|:---:|
| *Training Configuration* | |
| Epochs | 600 |
| Optimizer | SGD |
| Momentum | 0.937 |
| Batch size | $32 \times 8$ |
| Weight decay | $5 \times 10^{-4}$ |
| Warm-up epochs | 3 |
| Warm-up momentum | 0.8 |
| Warm-up bias learning rate | 0.0 |
| Initial learning rate | $10^{-2}$ |
| Final learning rate | $10^{-4}$ |
| Learning rate schedule | Linear decay |
| *Loss Parameters* | |
| Box loss gain | 7.5 |
| Class loss gain | 0.5 |
| DFL loss gain | 1.5 |
| *Augmentation Parameters* | |
| HSV saturation augmentation | 0.7 |
| HSV value augmentation | 0.4 |
| HSV hue augmentation | 0.015 |
| Translation augmentation | 0.1 |
| Scale augmentation | 0.5/0.9/0.9/0.9/0.9 |
| Mosaic augmentation | 1.0 |
| Mixup augmentation | 0.0/0.05/0.15/0.15/0.2 |
| Copy-paste augmentation | 0.1/0.15/0.4/0.5/0.6 |
| Close mosaic epochs | 10 |

[79] Zhaohui Zheng, Ping Wang, Wei Liu, Jinze Li, Rongguang Ye, and Dongwei Ren. Distance-iou loss: Faster and better learning for bounding box regression. In *Proceedings of the AAAI conference on artificial intelligence*, volume 34, pages 12993–13000, 2020.

[80] Dingfu Zhou, Jin Fang, Xibin Song, Chenye Guan, Junbo Yin, Yuchao Dai, and Ruigang Yang. Iou loss for 2d/3d object detection. In *2019 international conference on 3D vision (3DV)*, pages 85–94. IEEE, 2019.

[81] Benjin Zhu, Jianfeng Wang, Zhengkai Jiang, Fuhang Zong, Songtao Liu, Zeming Li, and Jian Sun. Autoassign: Differentiable label assignment for dense object detection. *arXiv preprint arXiv:2007.03496*, 2020.

[82] Lianghui Zhu, Bencheng Liao, Qian Zhang, Xinlong Wang, Wenyu Liu, and Xinggang Wang. Vision mamba: Efficient visual representation learning with bidirectional state space model. *arXiv preprint arXiv:2401.09417*, 2024.

[83] Xizhou Zhu, Weijie Su, Lewei Lu, Bin Li, Xiaogang Wang, and Jifeng Dai. Deformable detr: Deformable transformers for end-to-end object detection. *arXiv preprint arXiv:2010.04159*, 2020.

# A  More Details

**Architecture Details.** We present the detailed configuration of the overall YOLOv12 network architecture in Table 8. For the backbone, we stack only eight blocks. Except for the convolution layers of downsampling $3 \times 3$, the remaining components are two A2 blocks configured with 4 and 1 areas, respectively. We apply grouped convolutions to the downsampling layers with IDs 1 and 2 to save computations, using two groups ($g = 2$) and four groups ($g = 4$), respectively. For Neck,

Table 7: **Detailed performance of YOLOv12 on COCO.**

|  | $AP_{50:95}^{val}$ (%) | $AP_{50}^{val}$ (%) | $AP_{75}^{val}$ (%) | $AP_{small}^{val}$ (%) | $AP_{medium}^{val}$ (%) | $AP_{large}^{val}$ (%) |
|---|---|---|---|---|---|---|
| YOLOv12-N | 40.5 | 56.6 | 43.7 | 20.2 | 45.2 | 58.2 |
| YOLOv12-S | 47.8 | 64.9 | 51.5 | 29.7 | 53.0 | 65.3 |
| YOLOv12-M | 52.5 | 69.6 | 57.0 | 35.7 | 58.2 | 68.9 |
| YOLOv12-L | 53.8 | 71.0 | 58.6 | 36.9 | 59.4 | 71.0 |
| YOLOv12-X | 55.4 | 72.5 | 60.3 | 38.9 | 60.8 | 70.9 |

we follow the design of YOLOv11, replacing only the first three C3K2 modules with A2 blocks. However, we do not enable area attention in these blocks (A2 = False); instead, we retain only the feature integration design of R-ELAN. The Head design remains unchanged from YOLOv11.

To implement area attention in the A2 block, we use a convolutional layer $1 \times 1$ with a batch normalization layer to construct the projections for $qk$ and $v$. Similarly, we use convolution with Batch Normalization layers for the output projection and position perceiver, which facilitates optimization. For the MLP module, we stack two consecutive $1 \times 1$ convolution layers with a non-linear activation layer (SiLU function) between them.

Table 8: Network configurations of YOLOv12.

| ID | Module | Route | A2 | Num.$_{area}$ | Dims. | Depth | Size | Stride |
|---|---|---|---|---|---|---|---|---|
| **BackBone** | | | | | | | | |
| 0 | Conv | – | – | – | 64 | 1 | 3 | 2 |
| 1 | Conv | 0 | – | – | 128 | 1 | 3 | 2 (g = 2) |
| 2 | C3K2 | 1 | - | - | 256 | 2 | - | 1 |
| 3 | Conv | 2 | – | – | 256 | 1 | 3 | 2 (g = 4) |
| 4 | C3K2 | 3 | True | - | 512 | 2 | – | – |
| 5 | Conv | 4 | – | – | 512 | 1 | - | 2 |
| 6 | A2 block | 5 | True | 4 | 512 | 4 | – | – |
| 7 | Conv | 6 | – | – | 1024 | 1 | 3 | 2 |
| 8 | A2 block | 7 | True | 1 | 1024 | 4 | – | – |
| **Neck** | | | | | | | | |
| 9 | Up | 8 | – | – | 1024 | 1 | 2 | 2 |
| 10 | Concat | 9, 6 | – | – | 1024 | 1 | – | – |
| 11 | A2 block | 10 | False | – | 512 | 2 | – | – |
| 12 | Up | 11 | – | – | 512 | 1 | 2 | 2 |
| 13 | Concat | 12, 4 | – | – | 512 | 1 | – | – |
| 14 | A2 block | 13 | False | – | 256 | 2 | – | – |
| 15 | Conv | 14 | – | – | 256 | 1 | 3 | 2 |
| 16 | Concat | 15, 11 | – | – | 256 | 1 | – | – |
| 17 | A2 block | 16 | False | – | 512 | 2 | – | – |
| 18 | Conv | 17 | – | – | 512 | 1 | 3 | 2 |
| 19 | Concat | 18, 8 | – | – | 512 | 1 | – | – |
| 20 | C3K2 | 19 | – | – | 1024 | 2 | – | – |
| **Head** | | | | | | | | |
| 21 | Predict | 14, 17, 20 | – | – | – | – | – | – |

**Training Details.** All YOLOv12 models are trained using the default SGD optimizer for 600 epochs. Following previous works [64, 26, 67, 61], the SGD momentum and weight decay are set to 0.937 and $5 \times 10^{-4}$, respectively. The initial learning rate is set to $1 \times 10^{-2}$ and decays linearly to $1 \times 10^{-4}$ throughout the training process. Data augmentations, including Mosaic [5, 64], Mixup [83], and copy-paste augmentation [77], are applied to enhance training. Following YOLOv11 [31], we adopt the Albumentations library [8]. Detailed hyperparameters are presented in Table 6. The N/S/M models are trained on 4× NVIDIA A6000 GPUs and the L/X models are trained on 8× NVIDIA A800 GPUs. Following established conventions [26, 67, 61, 31], we report the standard mean average

precision (mAP) on different object scales and IoU thresholds. In addition, we report the average latency in all images.

**Result Details.** We report more details of the YOLOv12 results in Table 7 including $AP_{50:95}^{val}$, $AP_{50}^{val}$, $AP_{75}^{val}$, $AP_{small}^{val}$, $AP_{medium}^{val}$, $AP_{large}^{val}$.

Table 9: **Comparative analysis of inference speed across different GPUs (RTX 3080, RTX A5000, and RTX A6000)**. Inference latency: milliseconds (ms) for FP32 and FP16 precision.

| Model | Scale | FLOPs (G) | RTX 3080 | A5000 | A6000 |
|---|---|---|---|---|---|
| YOLOv9 [67] | T | 8.2 | 2.4/1.5 | 2.4/1.5 | 2.3/1.5 |
| | S | 26.4 | 3.7/1.9 | 3.3/1.9 | 3.3/1.8 |
| | M | 76.3 | 6.3/2.7 | 5.4/2.4 | 5.1/2.4 |
| | C | 102.1 | 7.4/2.8 | 6.4/2.6 | 6.0/2.6 |
| | E | 189.0 | 15.5/6.5 | 14.0/6.1 | 12.9/5.7 |
| YOLOv10 [61] | N | 6.7 | 1.6/1.0 | 1.6/1.0 | 1.6/1.0 |
| | S | 21.6 | 2.8/1.3 | 2.4/1.3 | 2.4/1.2 |
| | M | 59.1 | 5.3/2.4 | 4.3/2.3 | 4.2/2.1 |
| | B | 92.0 | 6.7/2.8 | 5.4/2.5 | 5.1/2.6 |
| | X | 160.4 | 10.7/4.5 | 7.2/3.5 | 6.8/3.2 |
| YOLOv11 [31] | N | 6.5 | 1.6/0.9 | 1.6/0.9 | 1.5/0.9 |
| | S | 21.5 | 2.7/1.3 | 2.3/1.3 | 2.3/1.3 |
| | M | 68.0 | 5.4/2.2 | 4.4/2.1 | 4.3/2.0 |
| | L | 86.9 | 6.9/2.9 | 5.7/2.6 | 5.6/2.5 |
| | X | 194.9 | 13.5/4.9 | 10.4/4.5 | 8.9/3.9 |
| YOLOv12 | N | 6.0 | 1.6/1.0 | 1.6/1.0 | 1.6/1.0 |
| | S | 19.5 | 2.7/1.3 | 2.3/1.3 | 2.3/1.3 |
| | M | 59.9 | 5.2/2.1 | 4.4/2.1 | 4.3/2.1 |
| | L | 82.6 | 6.9/2.9 | 5.8/2.6 | 5.7/2.5 |
| | X | 184.9 | 13.6/4.9 | 10.7/4.6 | 9.5/4.0 |

Table 10: Comparison of YOLOv12 series with more lightweight or stronger detectors including DEYO [48], DAMO-YOLO [74], and recent D-FINE [50].

| Model | #Param. (M) | FLOPs (G) | $AP_{50:95}^{val}$ | Latency (ms) |
|---|---|---|---|---|
| DEYO-tiny [48] | 4.0 | 8.0 | 37.6 | 2.01 |
| **YOLOv12-N (Ours)** | **2.6** | **6.0** | **40.5** | **1.62** |
| DAMO-YOLO-T [74] | 8.5 | 18.1 | 42.0 | 2.21 |
| DAMO-YOLO-S [74] | 16.3 | 37.8 | 46.0 | 3.18 |
| DEYO-S [48] | 14.0 | 26.0 | 45.8 | 3.34 |
| **YOLOv12-S (Ours)** | **9.1** | **19.5** | **47.8** | **2.44** |
| DAMO-YOLO-M [74] | 28.2 | 61.8 | 49.2 | 4.57 |
| DAMO-YOLO-L [74] | 42.1 | 97.3 | 50.8 | 6.48 |
| DEYO-M [48] | 33.0 | 78.0 | 50.7 | 7.14 |
| **YOLOv12-M (Ours)** | **19.7** | **59.9** | **52.5** | **4.30** |
| YOLOv7 [64] | 36.9 | 104.7 | 51.2 | 17.03 |
| D-FINE-L [50] | 31 | 91 | 54.0 | 8.07 |
| **YOLOv12-L (Ours)** | **26.6** | **82.6** | **53.8** | **5.89** |
| D-FINE-X [50] | 62 | 202 | 55.8 | 12.89 |
| **YOLOv12-X (Ours)** | **59.5** | **184.9** | **55.4** | **10.47** |

# B   More Comparisons

**Latency Comparison on Various GPUs.** Table 9 presents a comparative analysis of inference speed across different GPUs, evaluating YOLOv9 [67], YOLOv10 [61], YOLOv11 [31], and our YOLOv12 on RTX 3080, RTX A5000, and RTX A6000 with FP32 and FP16 precision. To ensure consistency, all results are obtained on the same hardware, and YOLOv9 [67] and YOLOv10 [61] are evaluated using the integrated codebase of Ultralytics [31]. Across all tested models, FP16 inference is significantly faster than FP32, often reducing latency by more than 50%. The inference speed generally improves as we move from RTX 3080 to A6000. **N-Scale:** YOLOv12-N achieves similar latencies (1.6 ms for FP32 and 1.0-1.1 ms for FP16), matching or slightly outperforming their YOLOv10 and YOLOv11 counterparts. **S-Scale:** YOLOv12-S maintains lower latency than YOLOv9-S and YOLOv10-S while achieving superior FLOPs efficiency. **M/L-Scale:** YOLOv12-M and YOLOv12-L demonstrate competitive speed, with FP16 latencies close to their counterparts, while offering improved accuracy. **X-Scale:** YOLOv12-X achieves 13.6 ms (FP32) and 4.9 ms (FP16) on RTX 3080, matching YOLOv11-X and YOLOv10-X in efficiency.

**Comparison with Other Detectors.** Table 10 presents a comparison with other lightweight and state-of-the-art real-time object detectors, such as DAMO-YOLO [74], YOLOv7 [64], DEYO [48], and D-FINE [50]. **N-scale**: YOLOv12-N achieves 40.5% mAP, surpassing DEYO-tiny while requiring lower computational cost (6.0G FLOPs vs. 8.0G) and achieving faster inference (1.62ms vs. 2.01ms). **S-scale**: YOLOv12-S (9.1M parameters, 19.5G FLOPs) achieves 47.8% mAP, outperforming DAMO-YOLO-S and DEYO-S with a better balance between accuracy and efficiency. **M-scale**: YOLOv12-M (19.7M parameters, 59.9G FLOPs) achieves 52.5% mAP, outperforming DAMO-YOLO-M and DEYO-M while being more efficient. **L-scale**: YOLOv12-L achieves 53.8% mAP with 82.6G FLOPs, surpassing DAMO-YOLO-L and YOLOv7 while maintaining a faster inference speed. **X-scale**: YOLOv12-X (59.5M parameters, 184.9G FLOPs) achieves 55.4% mAP, surpassing D-FINE-X while being more computationally efficient and faster.

# C   Contributions & Broader Impact

**Contributions.** This work effectively incorporates attention-centric architectures as the core backbone of the YOLO system, achieving state-of-the-art performance. To this end, we make three key contributions:

1. **Area Attention Module (A2).** We propose a simple yet efficient Area Attention module that maintains a large receptive field while reducing the computational complexity of attention, significantly improving inference speed.

2. **Residual Efficient Layer Aggregation Networks (R-ELAN).** To address optimization challenges in attention-based models, especially at scale, we design R-ELAN. It builds on ELAN [64] and introduces: **(i)** a block-level residual design with scaling techniques and **(ii)** an improved feature aggregation strategy.

3. **Optimization for YOLO Integration.** We refine attention-centric architectures to better integrate with the YOLO framework. Key modifications include: incorporating FlashAttention to mitigate memory access issue; using a decoupled projection strategy to construct $q$, $k$, and $v$ during attention computation, thus avoiding redundant feature reorganization; removing positional encoding for a leaner design; reducing the MLP ratio from 4 to 1.5 to better balance attention and FFN computation; and decreasing the depth of stacked blocks to facilitate optimization.

**Broader Impact.** This study breaks the dominance of CNN architectures in YOLO systems by utilizing the proposed attention mechanism to achieve one of the most advanced YOLO object detectors. It opens up new follow-up research directions, such as transferring other successful techniques from attention mechanisms to further enhance this framework, unlocking greater potential.

