# OpenReview forum: "YOLOv12: Attention-Centric Real-Time Object Detectors"
_NeurIPS.cc/2025/Conference — NeurIPS 2025 poster_

### Official Review · Reviewer_B1qM · 2025-06-19

**Clarity:** 3
**Significance:** 2
**Originality:** 3
**Rating:** 4
**Confidence:** 3

**Summary:**

This paper introduces YOLOv12, which integrates an attention-centric design into the YOLO framework. By combining area attention, R-ELAN, and architectural optimizations, YOLOv12 significantly improves accuracy and efficiency. Comprehensive experiments validate the effectiveness of the methods.

**Questions:**

1. Why are the networks compared between the left and right sub-graphs in Figure 4 inconsistent?
2. What training set is used  in the experiments? Is the experimental comparison fair?
3. I am confused about the proposed area attention. Why do others not support distortion-free rectangular inference but area attention does?

**Ethical Concerns:**

["NO or VERY MINOR ethics concerns only"]

**Limitations:**

Yes

**Quality:**

3

**Strengths And Weaknesses:**

Strengths
1. This paper proposes an attentioncentric YOLO framework that matches the speed of previous CNN-based ones while harnessing the performance benefits of attention mechanisms
2. The experiments and ablation studies are sufficient.
3. The proposed method seems simple and effective.

Weaknesses
1. As a YOLO series, it should be open sourced early.
2. Table 1 should indicate which models are based on CNN and which models are based on Attention.
3. It is recommended to add necessary code snippets to make it easier to understand area attention and R-ELAN.

---

> ### Author Rebuttal · Authors · 2025-07-26
>
> We appreciate the reviewer’s thoughtful review and positive comments about our study. In the following sections, we address the reviewer’s every concern and hope our responses address your concerns well. Further comments are welcomed.
>
> ***Q1：Open source the code early.***
>
> **A1:** ​Thank you for your valuable comment. YOLOv12 will definitely be open-sourced, including all code and model weights, ensuring that researchers and practitioners can easily access and build upon our work. We will also provide the code link in the final version.
>
> ***Q2: Table 1 should clarify CNN-based and Attention-based models.***
>
> **A2:** Thanks for your suggestion. We will indicate this in the final version. Specifically, among the YOLO series models, we are the only ones centered on attention, which strengthens the novelty of YOLOv12.
>
> ***Q3: Code snippets of area attention and R-ELAN.***
>
> **A3:** Thank you for your suggestion. We provide the pseudocodes of area attention and R-ELAN below. We will include them in the appendix and update them in the next version.
>
> **Area attention:**
>
> ```
> FUNCTION forward(x):
>     // Input shape: [B, C, H, W]
>     B, C, H, W = x.shape
>     N = H * W  // Total spatial positions
>     // Projections
>     qk = self.qk(x)          // Shape: [B, 2*all_head_dim, H, W]
>     v = self.v(x)            // Shape: [B, all_head_dim, H, W]
>     pp = self.pp(v)          // Positional perceiver
>
>     // Flatten to token sequence
>     qk = flatten(qk)  // [B, N, 2*all_head_dim]
>     v = flatten(v)    // [B, N, all_head_dim]
>
>     // Area-based reorganization
>     IF area > 1:
>         qk = reshape(qk, [B*area, N//area, 2*all_head_dim])
>         v = reshape(v, [B*area, N//area, all_head_dim])
>         UPDATE B = B * area, N = N // area
>
>     // Split query/key
>     q, k = split(qk, [all_head_dim, all_head_dim], dim=2)
>
>     // Attention computation
>     x = flash_attention(q, k, v)  // Output shape: [B, N, all_head_dim]
>
>     // Reverse area reorganization
>     IF area > 1:
>         x = reshape(x, [B//area, N*area, all_head_dim])
>         UPDATE B = B // area, N = N * area
>
>     // Spatial restoration
>     x = reshape(x, [B, H, W, all_head_dim])
>     x = permute(x, [0, 3, 1, 2])  // [B, all_head_dim, H, W]
>
>     // Final projection
>     RETURN self.proj(x + pp)
> ```
>
> **R-ELAN:**
>
> ```
> FUNCTION forward(x):
>     // Input shape: [B, C_in, H, W]
>
>     // Transition convolution
>     x_transition = transition(x)
>
>     current_input_for_block = x_transition
>
>     // Loop through the A2 blocks
>     FOR i FROM 1 TO num_A2_blocks:
>         // Store the input to the current A2 block before applying the block
>         input_to_current_block = current_input_for_block
>
>         // Apply the A2 block
>         A2_block_out = self.A2_blocks[i-1](current_input_for_block)
>
>         // Apply scaling to the input of the current block
>         scaled_input = self.scaling_layers[i-1](input_to_current_block)
>
>         // Add the scaled input to the block's output
>         current_input_for_block = A2_block_out + scaled_input
>
>     // After the loop, current_input_for_block holds the output of the last A2 block
>     last_A2_output = current_input_for_block
>
>     // with the output of the transition block.
>     concatenated_feature = concatenate([last_A2_output, x_transition], dim=CHANNEL_DIM)
>
>     // Final scaling and addition
>     scaled_concatenation = self.scaling_final(concatenated_feature) // Applying scaling
>     output = x + scaled_concatenation // Element-wise addition (residual connection)
>
>     RETURN output
> ```
>
> ***Q4: Sub-graphs inconsistent in Figure 4.***
>
> **A4:** Thank you for your careful observation. The main reason DETR-based methods are not included in the right figure is that these models have much slower CPU speeds compared to the YOLO series, which weakens the comparison between YOLOv12 and other YOLO models. To better compare models within the YOLO series, we did not include the DETR-based methods. We will clarify this in the revised version.
>
> ***Q5: Fair training settings and comparison?***
>
> **A5:** As we have emphasized, the training configuration of YOLOv12 is exactly the same as that of YOLO11, which ensures a fair comparison in the experiments.
>
> ***Q6: Why area attention support rectangular inference?***
>
> **A6:** Good question! The main reason area attention supports rectangular inference is that it does not rely on window design. Instead, it flattens the features into a one-dimensional vector and then uses reshape operations to split them into different areas. For example, suppose an image feature has a size of (4, 5). It is difficult to divide it into windows because windows of size (2, 2) or (3, 3) are both unsuitable. However, for area attention, we first flatten it into a one-dimensional vector with dimension (1, 20), then evenly reshape it into 4 or 5 parts to perform self-attention, thus supporting such rectangular inference. Our area attention is very simple and efficient to execute, which ensures its speed advantage.

---

### Official Review · Reviewer_XQeS · 2025-06-26

**Clarity:** 4
**Significance:** 4
**Originality:** 4
**Rating:** 5
**Confidence:** 5

**Summary:**

This paper presents YOLOv12, the latest version of the YOLO series, which significantly improves the accuracy of the YOLO family while maintaining comparable or even faster inference speed using an attention centric design. The contributions of YOLOv12 are multifaceted. For example, it introduces the Area Attention (A2) module, designs the Residual Efficient Layer Aggregation Networks (R-ELAN), and incorporates several technical optimizations, such as an innovative decoupled projection strategy for constructing Q, K, and V for attention. These innovations are original and, together, enable YOLOv12 to achieve a dominant latency-accuracy trade-off.

I believe this paper to be of high quality and clearly above the NeurIPS acceptance bar. I strongly recommend its acceptance.

**Questions:**

1. Will YOLOv12 support other tasks? It appears to be an innovation on architecture, which suggests that YOLOv12 could easily replace task heads to enable training for different tasks. If YOLOv12 can support more tasks, it would greatly expand its applicability and impact.
2. In line 169, the authors mention setting the MLP ratio to 2 for the N/S/M-scale models, and to 1.5 for the L/X-scale models. Why is there a difference in these settings?

**Ethical Concerns:**

["NO or VERY MINOR ethics concerns only"]

**Final Justification:**

I appreciate the authors' detailed response. The authors' rebuttal has taken care of my potential worries. This work makes a good contribution. My score remains unchanged.

**Limitations:**

The authors note that YOLOv12 relies on Flash Attention (FA), which limits its deployment on some devices. However, FA is likely to become core infrastructure in deep learning, with growing support across platforms. Thus, in my opinion, using FA in YOLO is a sensible and forward-looking choice.

**Quality:**

4

**Strengths And Weaknesses:**

Strengths:
1. This paper achieves strong latency-accuracy and FLOPs-accuracy trade-offs, outperforming previous YOLO models. It does not use any pre-trained weights, unlike DETR-based methods, which makes the performance even more impressive.
2. The proposed Area Attention is efficient and novel. It reduces attention complexity with just a simple flatten and reshape to build areas, avoiding previous complex designs. In addition, it also supports rectangular inference in YOLO, which prior models like Swin and CSWin do not—a clever and practical advantage.
3. The proposed R-ELAN is innovative and effectively mitigates the optimization challenges caused by attention in the YOLO framework. It offers a simple and generalizable solution to a common issue when adding attention to lightweight models.
4. Other designs, such as the decoupled projection strategy, also demonstrate innovation and effectively improve inference speed. Such engineering advances are particularly valuable for the YOLO series. The integration of Flash Attention is also interesting and timely—FA is likely to become a core infrastructure in deep learning, and adopting it in YOLO is a sensible and forward-looking decision.
5. This paper provides extensive ablation and diagnostic experiments on important designs, along with visualizations like those in Figure 5 to demonstrate YOLOv12’s advantages. The appendix also includes detailed information, such as architectural specifics.

Weaknesses:
1. I understand that DETR-based methods use pre-trained models, but YOLOv12’s performance is weaker than these methods—for example, D-FINE-X outperforms YOLOv12-X in accuracy (55.8 mAP vs. 55.5 mAP).
2. In Figure 3, CSPNet, ELAN, and R-ELAN are names of architectures or methods, while C3K2 refers to a module in YOLOv11. I believe these are not aligned consistently, and I suggest revising this for clarity.
3. I suggest the authors explain why the n-scale model of YOLO11 is faster than YOLOv12, while models of other scales are slower than YOLOv12.

---

> ### Author Rebuttal · Authors · 2025-07-26
>
> We appreciate this reviewer’s support for our work and the valuable comments provided in the review. We respond to your every concern and hope we have addressed them well. Your further comments are welcomed.
>
> ***Q1: Lower accuracy compared to D-FINE-X.***
>
> **A1**: We thank the reviewer for this observation. While it is true that D-FINE-X slightly outperforms YOLOv12-X in terms of mAP (55.8 vs. 55.5), our choice of YOLOv12 is deliberate and motivated by its suitability for real-time, resource-constrained applications. Unlike DETR-based methods, which typically rely on computationally expensive pretraining and large transformer backbones, YOLOv12 offers a more efficient alternative, achieving competitive accuracy with lower inference time (12.89 ms vs. 10.53 ms) and fewer computational resources. Additionally, it is worth noting that even without pretraining, YOLOv12 outperforms D-FINE on models of other scales.
>
> ***Q2: Clarify C3K2 for aligned consistently.***
>
> **A2**: Thanks for this suggestion. We agree that C3K2 is the name of a module rather than a method. Following your suggestion, we will replace C3K2 with GELAN for consistency.
>
>
>
> ***Q3: Why yolo11-n faster than yolov12-n while other scales not?***
>
> **A3:** Thanks for this careful observation. The fact that the n-scale YOLO11 model runs faster than YOLOv12, while the opposite is true for other scales, can be attributed to the extremely lightweight nature of the n-scale models, which have the smallest number of parameters and FLOPs. In this setting, the simpler architecture of YOLO11 incurs less overhead, resulting in slightly faster inference. As the model size increases (e.g., s, m, l, x), the architectural improvements in YOLOv12—such as better feature aggregation and a more efficient backbone—allow it to better utilize computational resources, leading to faster inference overall. As suggested, we will explain this in the revised paper.
>
>
>
> ***Q4: Support other tasks?***
>
> **A4:** Good question. Our YOLOv12 is indeed designed to support other tasks beyond object detection. For example, our experiments show that YOLOv12 can be easily adapted for instance segmentation and achieves state-of-the-art results. Specifically, YOLOv12-n/s/m/l/x achieve mAP50–95 (box) scores of ​39.9% / 47.5% / 52.4% / 54.0% / 55.2%, respectively, outperforming all corresponding versions of YOLO11.
>
>
>
> ***Q5: MLP ratio setting.***
>
> **A5:** This design is primarily intended to balance the parameter count and computational cost of YOLOv12, ensuring a fair comparison with previous versions such as YOLO11. By aligning the number of parameters and FLOPs of YOLOv12 with those of the corresponding YOLO11 versions, we enable a more equitable and meaningful evaluation.

---

> > ### Comment · Reviewer_XQeS · 2025-08-04
> >
> > I appreciate the authors' detailed response. The authors' rebuttal has taken care of my potential worries. This work makes a good contribution. My score remains unchanged.

---

> > > ### Author Response · Authors · 2025-08-04
> > > **Thanks**
> > >
> > > We are delighted that our response addressed your question. We appreciate your support for our work.

---

### Official Review · Reviewer_h5Ez · 2025-06-29

**Clarity:** 3
**Significance:** 3
**Originality:** 3
**Rating:** 4
**Confidence:** 5

**Summary:**

This paper proposes YOLOv12, an "attention centric" real-time object detection framework aimed at combining the expressive power of attention mechanisms with the efficiency of CNN architecture. The author introduces three improvements: (1) a local attention mechanism called Area Attention, which claims to reduce computational complexity while maintaining a long receptive field; (2) An improved feature aggregation structure R-ELAN for solving optimization instability problems; (3) Multiple attention structural modifications. Experiments have shown that it outperforms YOLOv10/11 and RT-DETR series on COCO datasets, achieving better trade-off in latency and accuracy.

**Questions:**

1. Please more clearly distinguish the essential difference between Area Attention and Local/Block Attention mechanism. At present, the essence is to divide the tokens into local groups for self-attention, which is a slight change of the existing methods.
2. Can we test the generalization ability of A2 in non-YOLO frameworks such as anchor-free detector or cascade head? Otherwise its generality is in doubt.
3. Is there a theoretical explanation for why the residual scaling + simplified attention combination provides stable training? Only empirical analysis is available.

**Ethical Concerns:**

["NO or VERY MINOR ethics concerns only"]

**Final Justification:**

Most of my concerns have been addressed, and I tend to keep my rating.

**Limitations:**

Refer to Weaknesses and Questions.

**Quality:**

3

**Strengths And Weaknesses:**

Strengths:
1. Strong actual performance: YOLOv12 significantly outperforms YOLOv10/11 in all model scales, especially in terms of latency/accuracy and FLOPs/mAP dimensions.
2. The engineering optimization is relatively comprehensive: combining multiple components such as FlashAttention, structural simplification, and decoupled projection, it demonstrates excellent system design capabilities.
3. Area Attention design is simple: Compared with window based attention, it supports any input resolution and is deployment friendly, especially suitable for common non square inputs in object detection.
4. Experimental adequacy: Detailed ablation experiments and speed evaluations are provided, covering GPU/CPU, FP16/FP32, and TensorRT environments.


Weaknesses:
1. The description of Area Attention in Figure 2 is not clear enough, and the intended meaning is not clear enough.
2. Suggest increasing the visualization comparison of receptive field results to enhance the reliability of the proposed method.
3. The essence of Area Attention is the local attention of token segmentation, which is similar to the early sliding window/local block attention or chunk-based attention, and there is no fundamental breakthrough in theory or form.
4. The structural adjustment of R-ELAN is a fine-tuning of ELAN, which is heuristic and lacks evaluation of sufficient theoretical analysis or generalization ability.
5. No significant improvement in modeling the nature of Transformers:
6. The authors claim "attention-centric" but mainly approximate attention to obtain CNN-like efficiency, which is essentially more like attention-inspired CNNS.
7. Despite removing positional encoding, location awareness is externalized to large-kernel conv, which is a common alternative in Transformers and not novel.

---

> ### Author Rebuttal · Authors · 2025-07-29
>
> We appreciate the reviewer’s constructive comments. Below, we address each of the reviewer’s concerns point by point. We hope our responses address all concerns, and we welcome any further feedback.
>
>
>
> ***Q1: The description of area attention is not clear enough, and the intended meaning is not clear enough.***
>
> ***A1:*** Thank you for your valuable feedback. We acknowledge that the current Figure 2 may not clearly convey the internal mechanism and advantages of the area attention (A2). As suggested, we will improve both the visual explanation and the accompanying textual description to clarify the intended meaning. Specifically, we plan to:
>
> 1. Add a diagram that illustrates the flatten → reshape → attention process step by step, to better demonstrate how A2 operates without relying on spatial windowing.
> 2. Clarify in the figure legend that ✓ / ✗ indicates support for rectangular inference, and explain key visual elements such as attention coverage.
> 3. Include a brief note explaining why window-based methods (e.g., Swin, CSWin) are incompatible with rectangular input sizes due to fixed window alignment, whereas A2 only requires the total token count to be divisible by $l$.
>
> We believe these changes will significantly enhance clarity and better communicate the core ideas of A2. We welcome any further suggestions.
>
>
>
>
> ***Q2: Suggestion on visualization comparison of receptive field results.***
>
> ***A2:*** Thank you for your insightful suggestion. We agree that comparing the receptive fields across different attention mechanisms is a valuable way to enhance the interpretability and reliability of our method. In the revised version, we will include a new figure that visualizes the attention maps and receptive field regions of the proposed A2 module. This comparison will illustrate how A2 enables large receptive fields at various values of $l$, particularly for non-square input resolutions, where other methods often struggle due to rigid window partitioning. We believe this visualization will further strengthen the empirical evidence for the effectiveness of A2 and help better convey its advantages.
>
>
>
>
> ***Q3: Area attention is similar to the window / local block / chunk-based attentions?***
>
> ***A3:*** Thanks for this thoughtful comment. We acknowledge that area attention (A2) shares conceptual similarities with prior local attention mechanisms. However, A2 introduces several key distinctions that, in our view, constitute a meaningful contribution in both design and practical effectiveness. Specifically:
>
> 1. Unlike fixed-size or fixed-shape designs typically used for image processing, A2 supports rectangular feature maps, allowing better adaptability to flexible input resolutions.
> 2. A2 introduces a decoupled projection that avoids redundant feature reorganization, enabling faster inference without compromising performance.
> 3. ​Empirically​, A2 consistently improves both accuracy and speed (Table 3), which fits well with the YOLO system.
>
> We believe that the novel design, efficient implementation, and demonstrated empirical effectiveness of A2 distinguish it from existing methods and justify its contribution.
>
>
>
>
> ***Q4: Theoretical analysis or generalization ability of R-ELAN.***
>
> ***A4:*** Thank you for this thoughtful comment. While R-ELAN builds upon ELAN, we emphasize that its design is not merely a heuristic fine-tuning, but rather a targeted architectural solution addressing specific limitations in ELAN—namely, the absence of residual connections, gradient blocking, and instability in deeper models with attention modules. We have conducted the following analysis and incorporated corresponding additions:
>
> 1. ​To demonstrate the generalization capability of R-ELAN​, we conducted ablation studies using R-ELAN in YOLO11-n/s models. The results show that models equipped with R-ELAN consistently achieve faster inference speeds and deliver comparable or even improved performance, while using fewer parameters and FLOPs. This supports the effectiveness and generalization of the R-ELAN design.
> 2. The introduction of residual connections and a scaling strategy effectively addresses the instability issues observed in ELAN-L and ELAN-X variants, particularly when integrated with attention modules. In addition, R-ELAN rethinks the feature aggregation strategy by avoiding concatenation of all intermediate outputs. This reduces redundancy and improves both parameter and memory efficiency—contributing to better convergence behavior and enhanced deployment practicality.
>
> We will highlight these aspects in the revised version, including the YOLO11 results, to clarify that R-ELAN is a general, transferable, and well-motivated architectural improvement.
>
> |Model|R-ELAN|FLOPs (G)|Param (M)|mAP|Lat.|
> |--|--|--|--|--|--|
> |N|✗|6.5|2.6|39.4|1.54|
> |N|✓|6.3|2.5|39.4|1.49|
> |S|✗|21.5|9.4|46.9|2.53|
> |S|✓|21.2|9.1|47.0|2.47|
>
>
>
>
> ***Q5: No significant improvement in modeling the nature of transformers.***
>
> ***A5:*** Thanks for pointing it out. We understand your concern regarding the depth of our contribution in relation to the fundamental modeling nature of Transformers. We respectfully clarify that area attention (A2) addresses several critical limitations of prior efficient attention mechanisms. Specifically, A2 supports native resolutions, freeing it from the limitations of window-based designs and enabling seamless integration with the YOLO framework. A2 introduces efficient operations; for example, it relies solely on flatten and reshape, allowing it to run efficiently. A2 also introduces a novel and elegant decoupled projection of q/k/v, which accelerates the framework by 10% without sacrificing accuracy. We believe these innovations empower YOLOv12 with a distinct advantage and differentiate it from other approaches.
>
>
>
>
>
> ***Q6: Attention-centric is essentially more like attention-inspired CNNs.***
>
> ***A6:*** We respectfully clarify that area attention (A2) is fundamentally an attention mechanism, not CNN-like. Like standard attention, A2 dynamically aggregates features based on token-to-token affinity within each area, while CNNs use static convolutional kernels. A2 retains the core attention paradigm of query-key-value projection and softmax normalization, only limiting computation spatially. Unlike CNNs’ fixed local receptive fields, A2’s adaptive affinity scores enable long-range dependencies across areas via stacked layers, maintaining Transformer behavior. We optimize attention for efficiency but do not approximate it with convolution. A2’s reshape-only partitioning avoids CNN priors, preserving attention’s dynamic nature.
>
>
>
>
>
> ***Q7: Removing positional encoding is a common alternative in transformer and not novel.***
>
> ***A7:*** Thanks for pointing it out. We acknowledge that the use of large-kernel convolution to externalize location awareness, in lieu of position encoding, has been explored in prior transformer variants. However, in our design, it serves as a minor architectural improvement rather than a core intellectual contribution. We note that this modification is well adapted to the area attention design, striking an excellent balance between accuracy and speed. As a result, it makes a valuable contribution to the overall performance of the framework.
>
>
>
>
> ***Q8: More clearly distinguish the essential difference between area attention and local / block attentions.***
>
> ***A8:*** Thanks for the suggestion! We clarify the essential distinctions between area attention and conventional local/block attention in the following three aspects. 1) Local/block attention requires a fixed window design to reduce complexity, while area attention uses flattened 1D segmentations on feature maps that are more flexible and support rectangular inference (important for the YOLO system). 2) Area attention is more efficient due to its elegant designs, such as decoupled projection and the requirement of only a flatten and a reshape operation, which is important for the efficiency of YOLOv12. 3) Area attention also showcases more powerful ability, as shown in Table 3.
>
>
>
>
> ***Q9: Test A2 in non-YOLO framework.***
>
> ***A9:*** Thank you for your suggestion. Following your advice, during the rebuttal, we replaced the third and fourth stages of the backbone module in the RT-DETRv2-S model with our A2 blocks. We adjusted the dimensions to ensure that the computational cost and parameter count remained approximately the same. Experimental results show that A2 achieved 48.4 mAP and 233 FPS, surpassing the original RT-DETRv2's 47.9 mAP and 217 FPS. This demonstrates the generalization ability of A2. We will include these results in the next version of the paper.
>
>
>
> ***Q10: A theoretical explanation for why the residual scaling + simplified attention combination provides stable training.***
>
> ***A10:*** Thank you for this constructive suggestion. The training stability primarily stems from the residual scaling mechanism, which addresses gradient pathologies in deep networks. Here is the theoretical rationale. In standard residual connections, the output is
>
> $$
> y = x + \mathcal{F}(x).
> $$
>
> When stacking attention blocks, gradient norms accumulate through the identity term:
>
> $$
> \frac{\partial\mathcal{L}}{\partial x}=\frac{\partial\mathcal{L}}{\partial y}\cdot\left(1+\frac{\partial\mathcal{F}(x)}{\partial x}\right).
> $$
>
> This can cause exponential gradient growth in deep chains. In R-ELAN's innovation, we scale residual connections by factor $0<\lambda <1$:
>
> $$
> y = x + \lambda \cdot \mathcal{F}(x).
> $$
>
> So that the gradient becomes:
>
> $$
> \frac{\partial\mathcal{L}}{\partial x}=\frac{\partial\mathcal{L}}{\partial y}\cdot\left(1 +\lambda \cdot\frac{\partial \mathcal{F}(x)}{\partial x}\right).
> $$
>
> The $\lambda$ term acts as a gradient dampener, constraining the maximum gradient norm to $O(1+\lambda)$ rather than $O(2^L)$ for $L$ layers. With residual scaling, $\lambda$ suppresses large gradient from attention, enabling safe integration. Thus, stability is enabled.

---

> > ### Comment · Reviewer_h5Ez · 2025-08-06
> >
> > Thanks for your response. Most of my concerns have been addressed, and I tend to keep my rating.

---

> > > ### Author Response · Authors · 2025-08-06
> > > **Thanks**
> > >
> > > We are very pleased that our response met your expectations. Thank you for your support of our work.

---

### Official Review · Reviewer_EVR3 · 2025-07-02

**Clarity:** 3
**Significance:** 3
**Originality:** 3
**Rating:** 4
**Confidence:** 4

**Summary:**

The paper proposes YOLOv12, an attention-centric real-time object detector that for the first time replaces the CNN-backbone tradition of the YOLO family with a pure attention design while matching—or surpassing—CNN speed.
 Key technical contributions are:
- Area Attention (A²): a simple “segment-wise” local self-attention that only needs one flatten → reshape pair. It preserves a large receptive field, supports any rectangular input (no window divisibility constraints), and runs faster than criss-cross, Swin or CSwin alternatives .
- R-ELAN backbone block: adds scaled residual shortcuts and a lighter aggregation strategy, stabilising training of deep attention stacks while cutting parameters/FLOPs .
- YOLO-specific refinements: FlashAttention integration, decoupled Q/K vs V projection (≈10 % speed-up) , removal of positional encodings, reduced MLP ratio (1.5) and shallower stages.

**Questions:**

1. A² Hyper-parameter l. You default to l = 4 or 8; how sensitive is accuracy and speed to this choice? A short plot would clarify robustness.
2. In Table 4, the authors show an ablation study in R-ELAN. Can you give another ablation study about using these efficient layer aggregation methods in previous models, like YOLOv11 for example?
3. In table 5, the authors show diagnostic studies about multiple methods. Can you use these methods in previous models and compare with your model? A table would be appreciated.

**Ethical Concerns:**

["NO or VERY MINOR ethics concerns only"]

**Limitations:**

Partially. The paper explicitly notes the FlashAttention hardware requirement , but societal impact is left as “NA”. I recommend adding a paragraph on misuse risks (e.g., mass surveillance) and dataset bias.

**Quality:**

3

**Strengths And Weaknesses:**

Strengths
1. Extensive evidence: state-of-the-art trade-offs vs. previous baselines; ablations isolate each component. Code-free tricks avoided, training from scratch.
2. Architecture diagrams, tables, diagnostic studies are easy to follow; checklist answers clarify compute and ethics.
3. Breaks the long-standing CNN dominance in YOLO by showing attention can equal latency; A² may benefit any dense detector needing rectangular inference.
4. A²’s extremely light implementation and decoupled-projection trick are novel; R-ELAN is a non-trivial improvement over ELAN.
Weaknesses
1. Speed advantage partly hinges on FlashAttention; without it, the gap shrinks (Table 5h)—thus impact depends on hardware adoption.
2. Use many known ingredients (FlashAttention, residual scaling); novelty is incremental relative to Swin/CSwin in spirit, though well-executed.\
3. Lack of theoretical analysis of A²'s superiority.

---

> ### Author Rebuttal · Authors · 2025-07-28
>
> We appreciate the reviewer's thoughtful feedback and support for our work. Below, we address the reviewer's concerns point by point. We hope our responses address all of your concerns, and we welcome any further comments.
>
> ***Q1: Speed advantage partly hinges on FlashAttention.***
>
> ***A1:*** Thank you for your insightful comment. We agree that FlashAttention contributes to the speed improvements observed in YOLOv12. However, we would like to emphasize that FlashAttention is increasingly becoming a standard infrastructure-level technique, with growing support across modern GPU hardware. In addition, YOLOv12 incorporates several other key design innovations to enhance efficiency, such as area attention and the decoupled projection of q/k/v, which also play significant roles in accelerating the model. Finally, as shown in Figure 4 (right), YOLOv12 maintains strong performance even on CPU devices without relying on FlashAttention.
>
> ***Q2: Use many known ingredients and novelty is incremental relative to Swin/CSwin in spirit, though well-executed.***
>
> ***A2:*** Thank you for this insightful comment. We agree that YOLOv12 adopts several well-established techniques. However, we would like to highlight that YOLOv12 also introduces multiple original and novel designs that play a crucial role in its performance. For example, the proposed area attention mechanism does not rely on fixed window patterns; instead, it employs a simple combination of flatten and reshape operations. More importantly, area attention naturally supports rectangular inference—a capability not available in Swin or CSWin. The comparison and advantages of area attention are clearly demonstrated in Table 3. Furthermore, the decoupled projection of q/k/v is a novel and elegant design that improves inference speed by 10% without compromising accuracy. We believe these innovations contribute meaningful novelty to the field and work synergistically to enable YOLOv12’s strong performance.
>
>
>
>
> ***Q3: Lack of theoretical analysis of A2's superiority.***
>
> ***A3:*** Thank you for your insightful comment. We agree that a formal theoretical analysis of the proposed area attention (A2) module would further strengthen our work. While our current focus is on empirical performance and practical efficiency, A2 is built upon clear design motivations that offer both conceptual novelty and practical benefits. Below, we outline its theoretical foundations and practical strengths:
>
> 1. **Receptive Field Control without Window Design.**
>    Unlike traditional local attention mechanisms (e.g., Swin, CSWin), which rely on fixed-size window partitioning, A2 avoids rigid window structures. It flattens a feature map of size \$(H, W)\$ into a sequence of $H \times W$ tokens and partitions them into $l$ equal-length segments. This strategy provides an elegant way to constrain the receptive field to approximately $\frac{1}{l}$ of the global context—achieving a similar effect to local attention—while remaining fully resolution-agnostic.
> 2. **Compatibility with Rectangular Inference.**
>    A2 only requires that the total number of tokens be divisible by $l$, without imposing any constraints on the spatial shape or aspect ratio of the input. This makes A2 inherently compatible with rectangular inference, which is standard in the YOLO evaluation protocol. In contrast, window-based methods often struggle under such conditions due to their reliance on fixed window configurations. A2's flexibility in this regard represents a practical advantage not offered by existing local attention designs.
> 3. **Implementation Simplicity and Efficiency.**
>    The A2 module consists of two simple tensor operations—flatten and ​reshape​—making it highly efficient and hardware-friendly. This simplicity translates into faster inference and facilitates seamless integration with real-time frameworks such as YOLO. Additionally, the use of decoupled projection of q/k/v reduces unnecessary feature reorganization in theory, further contributing to improved computational efficiency.
>
> We hope this explanation clarifies the motivation behind A2 and highlights its practical and conceptual merits, even in the absence of a formal theoretical analysis.
>
>
>
>
> ***Q4: How sensitive is accuracy and speed to the choice of $l$?***
>
> ***A4:*** Thank you for this valuable comment. We agree that the choice of the hyperparameter $l$ in A2 deserves clarification. During the rebuttal period, we conducted ablation studies using various values of $l$ (2, 4, 8, 16) in the third stage of the YOLOv12-n model. Specifically, these models achieve mAPs of 40.8%, 40.6%, 40.4%, and 40.0%, with corresponding inference times of 1.74 ms, 1.63 ms, 1.61 ms, and 1.58 ms on a T4 GPU, respectively. We will include a plot illustrating this trend in the appendix of the next version to better demonstrate the trade-off between speed and accuracy across different values of $l$.
>
>
>
>
> ***Q5: Use R-ELAN on YOLO11.***
>
> ***A5:*** Good suggestion! During the rebuttal period, we conducted ablation studies using the R-ELAN module in YOLO11-n and YOLO11-s models, and obtained the following results. The YOLO11-n/s models equipped with R-ELAN consistently achieved faster inference speeds and delivered comparable or even improved performance, while requiring fewer parameters and computational resources. These findings demonstrate the effectiveness and transferability of the R-ELAN design.
>
> | **Model** | **R-ELAN** | **FLOPs (G)** | **Param (M)** | **mAP** | **Lat.** |
> | --------------- | ---------------- | ------------------- | -------------------- | --------------- | ---------------- |
> | N              | ✗               | 6.5                 | 2.6                 | 39.4          | 1.54           |
> | N              | ✓               | 6.3                 | 2.5                 | 39.4          | **1.49**           |
> | S               | ✗               | 21.5                | 9.4                 | 46.9          | 2.53           |
> | S               | ✓               | 21.2                | 9.1                 | 47.0          | **2.47**           |
>
>
>
>
> ***Q6: Diagnostic studies using previous models and compare with YOLOv12.***
>
> ***A6:*** Good suggestion! Following your advice, we conducted additional experiments on the YOLOv10 and YOLO11 models (using n-scale variants) to validate the effectiveness of our proposed methods. Specifically, we performed diagnostic studies on several key components, including: (a) decoupled projection of q/k/v, (b) the position perceiver, (c) position embedding, and (d) FlashAttention (already reported in Table 5.f). The results are summarized below, from which we draw the following conclusions:
>
> 1. Decoupled projection is applicable to both YOLOv10 and YOLO11, and demonstrates speed improvements even though these models employ only a minimal amount of attention operations.
> 2. The proposed large-kernel position perceiver consistently improves model performance.
> 3. Position embedding does not improve performance and instead negatively impacts speed, which aligns with our findings in YOLOv12.
> 4. FlashAttention enhances the performance of YOLOv10 and YOLO11, achieving inference speeds comparable to YOLOv12.
>
> *Decoupled Projection*
>
> | Model   | Lat. (-DP) | Lat. | mAP |
> |---------|------------|------|-----|
> | YOLOv10 | 1.84       | 1.78 | 38.5 |
> | YOLO11  | 1.54       | 1.51 | 39.5 |
> | YOLOv12 | 1.71       | 1.63 | 40.6 |
>
> *Position Perceiver (PP)*
>
> | Model   | Lat./mAP (-PP) | Lat./mAP (+PP) |
> |---------|---------------|---------------|
> | YOLOv10 | 1.84 / 38.5   | 1.89 / 38.7   |
> | YOLO11  | 1.54 / 39.4   | 1.60 / 39.7   |
> | YOLOv12 | 1.57 / 40.3   | 1.63 / 40.6   |
>
> *Position Embedding*
>
> | Model          | Lat. | mAP |
> |----------------|------|-----|
> | YOLOv10 + RPE | 1.90 | 38.2 |
> | YOLOv10 + APE | 1.86 | 38.5 |
> | YOLO11 + RPE  | 1.64 | 39.2 |
> | YOLO11 + APE  | 1.57 | 39.4 |
> | YOLOv12       | 1.63 | 40.6 |
>
> *FlashAttention for YOLO-v10/11*
>
> | Model   | mAP | Lat. |
> |---------|-----|------|
> | YOLOv10 | 38.5 | 1.68 |
> | YOLO11  | 39.4 | 1.49 |
> | YOLOv12 | 40.6 | 1.63 |
>
> ***Q7: Societal impact: adding a paragraph on misuse risks.***
>
> ***A7:*** Thank you for raising this important point regarding societal impact. While our work focuses on real-time object detection with YOLOv12, primarily aimed at advancing efficient and accurate computer vision techniques, we acknowledge that such technologies could potentially be misused—for example, in mass surveillance or privacy-invading applications. In the next revision, we will include a dedicated paragraph discussing these potential misuse risks, as well as the importance of addressing dataset biases to promote fairness and responsible deployment. Furthermore, we emphasize that usage permissions should be strictly restricted in sensitive contexts to prevent abuse. We believe it is essential to raise awareness of the ethical implications of AI technologies alongside technical advancements.

---

> > ### Comment · Reviewer_EVR3 · 2025-08-04
> >
> > Thanks for your reply. Most of my concerns have been addressed, and I will keep my rating.

---

> > > ### Author Response · Authors · 2025-08-04
> > > **Thanks**
> > >
> > > We are happy that our response addressed your question. We appreciate your support for our work.

---

### Decision · Program_Chairs · 2025-09-17

**Decision:**

Accept (poster)

**Comment:**

This paper introduces YOLOv12, a novel, attention-centric architecture for real-time object detection that impressively matches the inference speed of prior CNN-based YOLO models while leveraging the performance benefits of attention. The core contributions, including the novel Area Attention mechanism that uniquely supports rectangular inference and an improved R-ELAN backbone for training stability, are well-motivated and empirically validated. The primary strengths of this work, as highlighted by all reviewers, are its state-of-the-art performance, achieving a superior accuracy-latency trade-off, and the thoroughness of its experimental validation and ablation studies. Initial weaknesses pointed out by reviewers included concerns about the novelty of some components, a dependency on FlashAttention, and the lack of theoretical justification for certain design choices. However, the authors provided an exemplary rebuttal, conducting new experiments that demonstrated the generalizability of their methods to other architectures (like YOLOv11 and RT-DETRv2), offering clear theoretical motivations, and clarifying all points of confusion. Given the paper's significant practical contributions to the widely-used YOLO family, its strong empirical results, and the authors' comprehensive engagement with the review process, AC recommends accept.